# Crystal Structure Reveals the Full Ras–Raf Interface and Advances Mechanistic Understanding of Raf Activation

**DOI:** 10.3390/biom11070996

**Published:** 2021-07-07

**Authors:** Trinity Cookis, Carla Mattos

**Affiliations:** Department of Chemistry and Chemical Biology, Northeastern University, Boston, MA 02115, USA; t.cookis@northeastern.edu

**Keywords:** Ras, Raf, Raf cystein-rich domain (CRD), Ras dimerization, HRas–CRaf-RBD_CRD crystal structure, Ras–Raf-RBD_CRD dimer simulations, allosteric connections, MAPK

## Abstract

Ras and Raf-kinase interact through the Ras-binding (RBD) and cysteine-rich domains (CRD) of Raf to signal through the mitogen-activated protein kinase pathway, yet the molecular mechanism leading to Raf activation has remained elusive. We present the 2.8 Å crystal structure of the HRas–CRaf-RBD_CRD complex showing the Ras–Raf interface as a continuous surface on Ras, as seen in the KRas–CRaf-RBD_CRD structure. In molecular dynamics simulations of a Ras dimer model formed through the α4–α5 interface, the CRD is dynamic and located between the two Ras protomers, poised for direct or allosteric modulation of functionally relevant regions of Ras and Raf. We propose a molecular model in which Ras binding is involved in the release of Raf autoinhibition while the Ras–Raf complex dimerizes to promote a platform for signal amplification, with Raf-CRD centrally located to impact regulation and function.

## 1. Introduction

Ras interacts with Raf kinase through Raf’s N-terminal Ras-binding (RBD) and cysteine-rich (CRD) domains to drive cell proliferation through the Ras–Raf–MEK–ERK mitogen-activated protein kinase (MAPK) pathway [1,2,3]. Mutations in Ras and Raf are major drivers of human cancers [4,5] and in spite of great efforts over the last 20 years, the mechanism for Ras-mediated activation of Raf remains elusive. This gap in our basic understanding of the pathway has limited our ability to explore novel strategies to develop drugs against Ras-driven cancers.

We previously solved crystal structures for wild-type Ras and its Q61L mutant bound to the Raf-RBD, revealing that this low-nanomolar-affinity complex occurs through electrostatic interactions at switch I (residues 30–40 in Ras) and that the oncogenic mutant RasQ61L has global impacts on the dynamics of the Ras–Raf complex [6,7]. Most recently, we showed that the Raf-RBD promotes dimerization of the Ras G domain [8], consistent with data supporting the requirement for Ras dimerization in the activation of the Ras–Raf–MEK–ERK signaling pathway [9,10,11,12]. Interestingly, both our HRas–CRaf-RBD (PDB ID 4G0N) and HRasQ61L/CRaf-RBD (PDB ID 4G3X) structures contain the Ras dimer with helices α4 and α5 at the interface, generated by taking the Ras–Raf-RBD complex in the asymmetric unit through a 2-fold crystallographic symmetry operation. Our molecular dynamics (MD) simulations of the dimer of the HRas–CRaf-RBD complex show that dimerization allosterically links the RBD sites for the scaffold protein Galectin-1 at the two ends of the dimer, suggesting a model in which the coupling of Ras–Raf dimers with Galectin dimers forms a signaling platform for kinetic proof reading and signal amplification [8]. This is consistent with the previously published observation that the Galectin-1 interaction with Raf is associated with an increase in HRas nanoclustering [13]. Galectins are part of a large family of β-galactoside-binding animal lectins that have highly homologous carbohydrate recognition domains and are multifunctional, with activities modulated by interactions with saccharides or peptides/proteins [14]. It is as scaffold proteins inside the cell that Galectin-1 and Galectin-3 have been associated with HRas and KRas activities, respectively [15,16]. The demonstration that this activity is through direct interaction between Galectin-1 dimers and CRaf-RBD was shown for HRas nanoclustering and signaling [13]. However, Galectin-3 readily dimerizes in the absence of glycans [17] and one can envision an analogous role in its signaling through KRas.

While the interaction between Ras and Raf-RBD has been well studied, the transformation potential of Raf requires interaction with Ras through both the Raf-RBD and Raf-CRD [2,18,19]. The recent crystal structure of KRas–CRaf_RBD_CRD (PDB ID 6XI7) has contributed to our understanding of the weaker micromolar affinity interaction between Ras and the CRD [20]. However, questions remain about a possible role of Ras dimerization and the function of the CRD in this context. The Raf-CRD requires coordination of two zinc ions and has been shown to interact with various phospholipids and supported lipid bilayers in vitro [3,21,22,23]. This interaction is driven primarily by electrostatic attraction between the positively charged CRD and negative phospholipid headgroups on the membrane [22]. The CRD membrane-binding role has further been studied by MD simulations and recently supported by NMR-guided characterization of nanodisc-bound KRas–CRaf-RBD_CRD complexes [24,25,26,27,28]. However, interaction of the CRD with the membrane has only been observed in vitro or in silico in the absence of other signaling components, with no evidence that this interaction is biologically relevant in cells.

In addition to the N-terminal Ras-binding conserved region (CR1) (residues 52–194, CRaf numbering), the three Raf isoforms (ARaf, BRaf, and CRaf) share two regions with high similarity that are connected through unstructured linker regions [5]. The second conserved region (CR2) (residues 254–269) contains a serine/threonine-rich domain with key phosphorylation sites [29,30,31] and the third conserved region (CR3) comprises the C-terminal kinase domain (residues 349–609) (Figure 1a). Prior to its recruitment to the membrane and interaction with Ras, Raf is held in an autoinhibited state through the aid of 14-3-3 scaffold proteins that bind motifs located in the CR2 and CR3 regions [32,33,34]. Recruitment of Raf to the membrane through Ras results in dimerization of the Raf kinase domain, leading to its activation and subsequent phosphorylation of MEK to propagate signaling [35,36]. Recent cryo-EM structures of inactive and active BRaf/14-3-3 and BRaf/MEK1/14-3-3 complexes have shed light into this mechanism revealing a critical role for the Raf-CRD in stabilizing the autoinhibited state [32,37,38]. It has been noted that the Raf-RBD is exposed in the complex with Ras, allowing its interaction with Ras to displace the CRD from the kinase/14-3-3 complex [20]. From there, the proposed model for Ras activation of Raf is built on the premise that the CRD must both bind Ras and insert into the membrane upon release from its autoinhibition role, leading to a signaling state in which Ras is monomeric on the membrane [20].

Here, we present the crystal structure of HRas bound to a construct containing both the CRaf-RBD and CRD at 2.8 Å resolution (PDB ID 7JHP) (Figure 1b,c and Appendix A) and build on our recently proposed model for Ras activation of Raf [8]. Our model is based on the discovery that, on supported lipid bilayers, Ras dimerizes upon binding to Raf, and allows for the possibility that previously observed insertion of the CRD into the membrane is an artifact of experimental conditions. Our crystal structure of the HRas–CRaf-RBD_CRD is nearly identical to that of the recently published KRas–CRaf-RBD_CRD structure [20], adding to evidence that the Ras isoforms activate Raf kinase through a common mechanism [8]. This structure reveals that the Raf-CRD interacts with Ras residues at a surface contiguous to the Raf-RBD along β2, stretching to loop 3 at the interswitch region. As the dimer is not present in our crystals (Appendix A), we used this structure to build a model of the HRas–CRaf-RBD_CRD dimer based on our HRas–CRaf-RBD structure, which places the Raf-CRD at the base of the two Ras protomers opposite the C-terminal end of α5 and the HVR, facing away from the expected location of the membrane. In this model, the CRD is poised to have an impact on both the Ras active site and allosteric connections in the signaling complex. Significantly, although not noted in the publication [20], this dimer is present through a two-fold crystallographic symmetry axis in the higher resolution crystal structure of the KRas–CRaf-RBD_CRD complex (PDB ID 6XI7) (Appendix A), consistent with our model and the analysis presented here. Based on our crystal structure and recent developments in the literature, we propose an alternate mechanism in which binding of Raf-RBD promotes dimerization of the Ras–Raf-RBD_CRD complex, with the CRD positioned at the base between the two Ras molecules in the dimer, where it has the potential to allosterically affect regulation and signaling output.

## 2. Materials and Methods

### 2.1. Protein Expression/Purification

The HRas–CRaf-RBD_CRD complex was crystallized using the HRasR97C mutant. The HRasR97C construct containing residues 1–166 (henceforth referred to as HRas) was generated by PCR site-directed mutagenesis from the wild-type HRas construct similarly truncated. HRas protein was expressed, purified, and GDP was exchanged for the non-hydrolyzable GTP analog GppNHp, as previously described [39]. The GB1_Raf-RBD-CRD (52–184) construct, as previously published [7,40], was extended utilizing PCR site-directed mutagenesis to generate the construct of CRaf-containing residues 52–187 used for the experiments described in this paper. Cys 184 coordinates a Zn^2+^ ion and truncation at this position led to a disordered CRD in our previously published structure [7]. CRaf protein was transformed into *Escherichia coli* BL21(DE3) cells and grown at 37 °C at 220 rpm in the presence of 50 mg/L ampicillin. Expression was induced in the presence of 20 μM ZnCl_2_ at an OD_600_ of 0.6–0.8 with 0.5 mM IPTG. The temperature was lowered to 32 °C and cells were harvested after 5 h by centrifugation.

Cells were resuspended in 50 mM Tris-HCl pH 7.4, 500 mM NaCl, 1 mM BME, 5% *v*/*v* glycerol, and 20 μM ZnCl_2_ in the presence of 1 mg/mL leupeptin, 1 mg/mL pepsatin, and 12 mg bezamidine. Cells were lysed by sonication and the insoluble fraction was separated by centrifugation at 13,500 rpm, 4 °C for 30 min. The supernatant was syringe filtered through a 0.45 μm membrane before performing Ni NTA affinity chromatography (HisTrap HP, GE Lifesciences). CRaf protein was eluted with 200 mM imidazole. Fractions were pooled in the presence of 1 molar equivalent HRas and concentrated to 2 mL. Protein was dialyzed overnight into cleavage buffer (20 mM Tris-HCl pH 8.1, 100 mM NaCl, 1 mM DTT, 25 mM CaCl_2_, 5 mM MgCl_2_, 20 μM ZnCl_2,_ 5% *v*/*v* glycerol) at 4 °C. Ten units of thrombin were added per mg of CRaf_52-187 and incubated for 18 h at room temperature. The cleaved complex was further purified by size exclusion chromatography (16/60 Sephacryl S100, GE Lifesciences).

### 2.2. Protein Crystallization, Data Collection, and Refinement

The HRas–CRaf_52-187 complex containing RBD and CRD was concentrated to 7 mg/mL for crystallization trials. Crystals of the complex were obtained by sitting-drop vapor diffusion with a reservoir solution of 0.1 M ammonium acetate, 0.1 M Bis-Tris pH 5.5, and 17% *w*/*v* PEG 10K. Drops were prepared with 1 μL protein and 1 μL reservoir solution and crystals were grown at 18 °C over ten weeks. Data collection was performed on a home source MicroMax007HF with a Cu^2+^ anode, tungsten filament, and R-AxisIV^2+^ detector from Rigaku. Data were indexed, integrated, and scaled with the HKL3000 software package [40]. Molecular replacement was performed in PHENIX [41] utilizing the HRas–CRaf-RBD crystal structure (PDB ID 4G0N) as the search model. After correct placement of Ras and Raf-RBD molecules, an additional search was performed for the Raf-CRD utilizing the Raf-CRD NMR structure (PDB ID 1FAR). Further structure refinement was performed with PHENIX [41] and COOT [42].

### 2.3. Model Preparation for Molecular Dynamics Simulations

The monomer of the HRas–CRaf-RBD_CRD complex was prepared by substituting the fully ordered wild-type HRas G domain (PDB ID 3K8Y) for the RasR97C G domain in the HRasR97C/CRaf-RBD_CRD crystal structure. Loop 4 of the Raf-RBD (residues 103–109) was modelled using the coordinates of the Raps/Raf-RBD crystal structure (PDB ID 1C1Y). The Raf-RBD_CRD linker region and unmodelled side chains were built using guidance from the 2F_o_-F_c_ electron density map contoured at low sigma levels.

To obtain the dimer model with the α4–α5 interface frequently observed in crystal structures, the dimer generated by 2-fold crystallographic symmetry from our structure of HRas–CRaf-RBD (PDB ID 4G0N) was used as a template for superposition of two HRas–Raf-RBD_CRD complexes to generate the dimer containing the CRD.

The six cysteines located in the CRaf-CRD for all systems were patched with CYN (deprotonated Cys residues) for stable coordination of the two Raf-CRD zinc ions throughout the simulations. All histidines in both HRas and CRaf molecules were in the uncharged HSE form.

### 2.4. Molecular Dynamics Simulations

Each system was solvated in a TIP3P water box and sodium and chloride ions were added to a final concentration of 150 mM to neutralize the system. Each system was minimized for 5000 steps and gradually heated from 50 to 250 K prior to the production runs. Production runs were performed at constant temperature and pressure of 300 K and 1.01325 bar with periodic boundary conditions. The first 30 ns of each simulation had a time step of 1 fs and the remaining simulation time was performed with a time step of 2 fs. Simulations were performed with the NAMD package [43] and CHARMM27 force field [44]. The Particle Mesh Ewald method with a grid size of 1 Å was used to calculate long-range electrostatics. Simulations for the HRas–CRaf-RBD_CRD dimer model were performed in three independent runs, each for a duration of 350 ns, for a total of 1.05 µs.

### 2.5. Trajectory Analysis

Dynamical network analysis was performed using the Carma [45] and Catdcd software packages. Each protein residue within the simulation is assigned a spherical node centered on its alpha carbon. Edges are drawn between nodes that stay within 4.5 Å of each other for at least 75% of the trajectory and are weighted utilizing pairwise cross correlation data. Optimal and suboptimal path calculations can also be used to identify residues important in allosteric communication. The optimal path is calculated as the path containing the fewest number of edges connecting user specified “source” and “sink” nodes. Nodes occurring most frequently in the calculated optimal and suboptimal paths are important for allosteric communication between the two specified “source” and “sink” nodes [46].

Root-mean square deviation (RMSD) and root-mean square fluctuation (RMSF) calculations were performed using the Bio3d software package [47]. Distance calculations were performed in VMD [48] by sourcing the distance.tcl script that can be found in the VMD script library.

### 2.6. Protein Interfaces Surfaces and Assemblies (PISA)

PISA is a program for analyses of PDB structures including surfaces associated with crystal contacts to determine the likelihood that an interface is an artifact of crystal packing [49]. For this, it uses the interface free-energy Δ^i^G *p*-value as a measure of whether the interface hydrophobicity is consistent with the average of biologically relevant interfaces. A *p*-value < 0.5 means that the interface is more hydrophobic than the average and is likely to be biologically relevant, whereas a *p*-value > 0.5 indicates that the surface is probably an artifact of crystallization. Further detail can be found at the European Bioinformatics Institute webpage: http://www.ebi.ac.uk/pdbe/prot_int/pistart.html (accessed on 1 June 2021).

## 3. Results

### 3.1. Overall Structure of Ras–Raf-RBD_CRD Complex

The Raf-RBD and CRD interfaces are contiguous surfaces spanning Ras residues 23 through 48, including the C-terminal end of the Ras helix α1, switch I, β2 and loop 3 (Figure 1b). The Raf-RBD interacts with Ras at switch I and the N-terminal portion of β2 (residues 27–41), as previously described [7], and the short linker connecting the Raf-RBD and Raf-CRD (residues 133–137) is partially disordered in our structure, with poor electron density for residues 134–136. The Raf-CRD interaction involves hydrogen bonds and hydrophobic contacts with C-terminal residues in Ras α1 (residues L23-N26) and β2, extending to loop 3 in the interswitch region (residues K42-G48), with direct and allosteric connections to α5 (Figure 1c). The Raf-CRD interface involves residues near the C-terminal end of the CRD in the stretch from F172 to T182, and nearby N-terminal residues H139, F141, R143, as well as F163. This interface includes the helix adjacent to the first Zn^2+^-binding site and extends toward the C-terminus, where the second Zn^2+^ ion is coordinated by residues C184 as well as by the interface residue H139, linking the N and C-terminal ends of the Raf-CRD. The position of the CRD observed in our crystal structure is as recently reported [20] and consistent with published NMR chemical shift perturbation (CSP) data (Figure 1c and Appendix A), and with Ras N26G and V45E mutations shown to disrupt the Ras–Raf-CRD interaction [19,24]. Given its close contact with the interswitch loop 3 on Ras, the Raf-CRD is allosterically connected to α5 through salt bridges that form between Ras D47 and E49 on loop 3 and R161 and R164 on α5 (Figure 1c). This binding mode of the Raf-CRD to Ras is distinct from that described in the NMR data-driven nanodisc-bound KRas–CRaf-RBD_CRD complexes (Appendix A), where the Ras–Raf-CRD interaction was modelled utilizing chemical shift perturbation data that are also consistent with our model (Appendix A) and paramagnetic relaxation enhancement (PRE) data with placement of the probe at a cysteine engineered in place of Q43 [24], a residue with multiple interactions at the Ras–Raf-CRD interface (Figure 1c). Only the RBD-CRD configuration in the crystal structure is consistent with the expected location of the Raf-RBD when superimposed on the cryo-EM structure for the inactive BRaf/MEK1/14-3-3 complex (PDB ID 6NYB), in which the Raf-CRD is sandwiched between the Raf-kinase domain and 14-3-3 dimer and the Raf-RBD is solvent exposed, absent from the final cryo-EM reconstruction [32] (Appendix A).

### 3.2. Raf-CRD Links to the Active Site through Loop 8 across the Dimer Interface

To study Ras–Raf in the context of the dimer and determine possible roles for the CRD, we modelled the HRas–CRaf-RBD_CRD dimer utilizing a two-fold symmetry axis conserved across many Ras crystal structures, including our structure of the HRas–CRaf-RBD complex (PDB ID 4G0N) containing the α4–α5 dimer interface [7,50,51]. In this model, the Raf-CRD is positioned for interaction with both Ras protomers in the Ras dimer, bridging the interswitch region of one to loop 8 of the other at the base of the Ras dimerization interface (Figure 2a), as is found in the recently published KRas–CRaf-RBD_CRD crystal structure containing the dimer (PDB ID 6XI7) [20]. We performed three independent replicates of molecular dynamics (MD) simulations, each for a duration of 350 ns, totaling just over 1 µs. The model equilibrated quickly and was stable throughout the trajectories, with average RMSD near 2.5 Å from the starting model in each case (Appendix A). The simulations show greater fluctuations for Raf-CRD than observed for the rest of the complex (Appendix A), with sampling of interactions seen in the KRas–CRaf-RBD_CRD crystals where the dimer is present.

In the starting model for simulations of the HRas–CRaf-RBD_CRD dimer, Raf-CRD residues R143, N161, and E174 are within interacting distance of loop 8 residue R123, the nucleotide-binding residue D119, and Q150 at loop 10 immediately preceding α5 in the opposing Ras protomer. Ras D119 is part of the NKxD motif (residues 116–119), and directly binds the guanine base in the Ras active site (Figure 2b,c). This motif is followed by loop 8, containing residue R123 which makes a salt bridge interaction with E143 of the ExSAK motif (residues 143–147), linking two highly conserved regions important for active site stability [52]. The Ras R123–E143 salt bridge remains present throughout the simulations, even as R143 of the Raf-CRD moves to interact with Ras E143 (Figure 2b,c, Appendix A (replicate 1), Appendix A (replicate 2) and Appendix A (replicate 3)), contributing to the interface with the opposing Ras promoter. Raf-CRD residue N161 samples various states in which it can make contacts with the side chains of Ras T148 and Q150 in loop 10 (Figure 2b,c, Appendix A) or with the backbone of loop 8 residues C118, D119, and L120 (Figure 2b,c, Appendix A). Ras Q150 also interacts with Raf-CRD E174 and H175 (Figure 2b,c, Appendix A). Raf-CRD T145 is also observed to interact with the loop 8 residues D119 and L120 (Figure 2b,c). Various interactions revealed in the simulations, although not initially present in the model built from our structure of the HRas–CRaf-RBD_CRD in which the dimer is not present (PDB ID 7JHP), are observed in the recently published structure of the KRas–CRaf-RBD_CRD dimer (PDB ID 6XI7). Most notably, the interactions between the Raf-CRD and both the glutamate and lysine of the ExSAK nucleotide-binding motif are observed both in our simulations and KRas–CRaf-RBD_CRD dimer structure (PDB ID 6XI7). It is clear that the interactions between the CRD and Ras across the dimer interface are dynamic, with the potential of stabilizing different sets of interactions depending on the state of the signaling complex. The dynamic ability of the two Raf-CRDs to interact with the opposing Ras molecule provides a means to introduce some local asymmetry into the Ras–Raf dimer complex (Figure 2b,c, Appendix A). There are several isoform-specific residues in this region of Ras (Appendix A) [52], thus increased dynamics of the Raf-CRD may be a key feature allowing the three Ras isoforms to have specific interactions with each of the Raf proteins [53]. Furthermore, the asymmetry and flexibility that we observe in the dimer may facilitate a given Ras isoform to accommodate heterodimers of Raf [54], as CRaf residues N161, E174, and H175 are isoform-specific and correspond to Q257, Q270, and R271 in BRaf (Appendix A). Overall, in the Ras–Raf dimer, the Raf-CRD has access to the Ras dimerization interface through its interactions with loop 3 and α5, and to the Ras active site through loop 8, poised to affect key allosteric connections involved in the regulation of Ras [55].

### 3.3. Allosteric Communication across the Ras–Raf-RBD_CRD Dimer Complex

We recently demonstrated that the Raf-RBD promotes Ras dimerization and used dynamical network analysis and optimal/suboptimal path calculations based on MD trajectories to identify allosteric connections linking both Ras dimerization and Raf-RBD interfaces in the HRas–CRaf-RBD dimer [8]. We use the same type of analysis here to identify pathways of allosteric connections in the HRas–CRaf-RBD_CRD dimer and compare them with those we found in the absence of the CRD. Briefly, a network is constructed of edges drawn between residues that stay within 4.5 Å of each other for at least 75% of the simulation time and edges are weighted as a function of pairwise correlations [46]. The resulting connectivity network allows for the calculation of allosteric paths through residues that link distant regions of the protein or protein complex, with the optimal path representing the shortest sum of edge weights between two residues and suboptimal paths deviating from the optimal path within a user-specified threshold. In our previous work, we studied the paths linking the two Raf-RBD residues D113 in the dimer of the HRas–CRaf-RBD complex due to its positioning, along with Raf-RBD D117, in the Raf-RBD pocket for the scaffold protein Galectin-1, critical for the activation of the MAPK pathway [13]. We observed strong allosteric linkages between the D113 residues extending 85 Å across the Ras–Raf-RBD dimer complex, and identified residues D47, located in loop 3 of one Ras protomer, and E143, in β6 of the other, as a critical point of information transfer across the Ras dimerization interface [8].

It is intriguing that the Raf-CRD is located at the base of the Ras–Raf dimer to interact with both loop 3 of one Ras molecule and residue E143 of the other, therefore in contact with residues identified as critical for extending allosteric communication across the Ras dimerization interface [8]. Dynamical network analysis was performed for each simulation replicate to identify residues with correlated motions within the dimer of the HRas–CRaf-RBD_CRD complex and involved in intermolecular information transfer across the complex [56,57]. The edge weights, determined as a function of pairwise correlations and used for optimal and suboptimal path calculations, for residues that cross the Ras–Raf-CRD’, Ras–Raf-RBD_CRD and Ras–Ras’ interfaces in the three MD simulation replicates, demonstrate the variability across both Ras–Raf-CRD interfaces in the global analysis (Appendix A), consistent with the high RMSF associated with the Raf-CRD (Appendix A). Due to longer simulation run times and the dynamic nature of the Ras–Raf-CRD interactions across the dimer interface (Appendix A), dynamical network analysis was also performed by dividing each simulation into seven segments of 50 ns each in order to capture the diversity in communication extending across the complex (Figure 3a). Analysis of optimal and suboptimal path calculations between the two D113 residues of each Raf-RBD in the networks constructed over 350 ns of simulation time, are consistent with our HRas–CRaf-RBD dimer simulations [8], demonstrating the conservation of intermolecular information transfer involving the interswitch region of one Ras protomer and β6 of the other across the Ras dimerization interface (Figure 3a,b, orange path). Additionally, by constructing networks for smaller simulation segments, we capture four distinct modes of intermolecular information transfer that are consistent across the three replicates (Figure 3a and Appendix A) (Appendix A). The first mode of information transfer involves the interswitch region and β6 as described above (Figure 3b, orange path). The second mode involves α5 and β6 as described for KRas–CRaf-RBD dimer simulations modelled on the membrane [8] (Figure 3b, red path). The remaining two means for information transfer directly involve the Raf-CRD, with one utilizing the Raf-CRD’s close association with loop 8, creating paths that wrap around the Ras active site (Figure 3c); and the other alternatively using the interconnectivity between the Raf-CRD and interswitch region to link the two Ras protomers again through loop 3 and β6 (Figure 3d).

The allosteric connections linking residues D113 at the Galectin-binding pocket of each Raf-RBD molecule in the dimer, which also includes residue D117 [13] (Appendix A), suggest a model where Ras–Raf-RBD_CRD dimers couple with Galectin dimers to form a higher-order macromolecular platform involved in signal amplification and kinetic proofreading [8] (Figure 3e), similar to that described for the LAT/Grb2/SOS system [58]. This model is consistent with the functional importance of Galectins 1 and 3 for signaling through the MAPK pathway [15,16,59,60] and with results from single molecule tracking experiments correlating immobile species of Ras observed in live cells to active signaling through the Ras–Raf–MEK–ERK pathway [61,62,63].

Our molecular dynamics simulation data suggest that the interconnectivity between the Raf-CRD and the interswitch region of one Ras molecule and loop 8 of the other, in the context of its dynamic nature, optimally positions the Raf-CRD to diversify the paths of allosteric communication across the Ras–Raf complex linking the Galectin-binding sites at the two ends of the dimer, not only by further bridging the two Ras molecules in the Ras dimer, but also serving as a link between the Raf-RBD and second Ras molecule in the complex, with a direct connection to the Ras active site. Simulations of the KRas–CRaf-RBD dimer complex modelled on the membrane demonstrate that this complex is flexible at the interface [8]. The Raf-CRD may accommodate the complex’s dynamic nature by providing alternative routes of communication between the two Galectin-binding regions at opposite ends of the dimer. The prominence of each of the alternate allosteric paths facilitated by the CRD may be modulated by isoform-specific residues and sensitive to oncogenic mutations in the active site.

## 4. Discussion

Ras nanoclustering on the membrane modulates Ras activation of MAPK signaling, but the significance of Ras dimers as functional signaling units is still not fully resolved in the literature [64]. Weak binding is predicted for Ras dimers in the absence of other factors [25,64,65,66,67], putting in question its viability as a robust signaling unit. However, we have recently shown that Raf-RBD promotes robust dimerization of Ras on supported lipid bilayers even at very low surface densities, in contrast with Ras in the absence of Raf-RBD, for which no dimers can be detected [8,68]. This is consistent with observation of dimeric Ras in live cells [12] and with experiments that suggest a requirement for Ras dimerization in the activation of Raf [9,10,11,12].

The recently published crystal structures of the KRas–CRaf-RBD_CRD complex, each with one complex monomer in the asymmetric unit in two different crystal forms (PDB IDs 6XI7 and 6XHB), led authors to suggest a model in which Ras acts as a monomer in the complex with Raf [20], building on an emerging model for Ras activation of Raf [69]. Here, the kinase domains of two nearby complexes dimerize to activate the signal without the need for the Ras molecules to dimerize. This model is consistent with years of research showing interaction of the CRD with the membrane in vitro and in silico [3,21,22,23,24,25,26,27,28]. Interestingly, one of the published crystal forms (PDB ID 6XI7) has the dimer in the crystal with a structure that converges on our model of the dimer built from the crystal structure of the HRas–CRaf-RBD_CRD (PDB ID 7JHP). Here, we offer an alternative model of Raf activation in which Ras dimerization upon binding Raf-RBD is a key step, with placement of the CRD at the base of the dimer interface, allosterically connecting elements of α5, the interswitch region and two highly conserved elements of the nucleotide-binding site: the NKxD and ExSAK motifs. Within the bigger picture of signaling through Ras–Raf–MEK–ERK, the primary function of Ras in our model is not in promoting Raf dimerization *per se*, but in providing an essential building block for a signaling platform in which Ras–Raf dimers couple with Galectin dimers, leading to synchronized activation of multiple Raf kinase dimers, stabilized by 14-3-3, for signal amplification (Figure 3e). Binding of short oligosaccharides to Galectin-1 has recently been shown to promote allosteric connections to the dimer interface [70]. Given that the proposed site for Raf-RBD partially overlaps the glycan ligand-binding site [13], the binding of HRas–CRaf-RBD_CRD to Galectin-1 as we propose could result in allosteric connections across the entire signaling platform. Galectin-3, known to be monomeric when bound to glycans, readily self-dimerizes in the apo form [17], such that one could envision an analogous KRas–Raf-RBD–Galectin-3 signaling platform. The Raf-CRD in this context is centrally and flexibly located to affect regulation both through diversifying the paths of allosteric connections in the signaling complex and possibly affecting active site function, such as GTP hydrolysis, through its proximity to the nucleotide-binding pocket.

We recognize that the specific interface through which Ras dimerizes has not yet been unequivocally determined [64]. However, both the HRas–CRaf-RBD (PDB ID 4G0N) and the KRas–CRaf-RBD_CRD (PDB ID 6XI7) dimer structures with the α4–α5 Ras dimer interface, as well as the many studies supporting Ras dimerization through this interface [8,9,50,51,71], point to our proposed model for the dimer. Given this dimer, we can hypothesize its placement with respect to the membrane in a signaling context. The evidence supporting an approximately perpendicular orientation of α3, α4 and α5 with respect to the membrane [8,50], and the fact that Ras is tethered to the membrane through the HVR following the C-terminal end of α5, places the dimer with the Raf-CRD away from the membrane. This dimer model, which is derived from crystal structures and remains to be confirmed experimentally on the membrane, would preclude the CRD insertion into the membrane. We thus question the presumed membrane-binding function of the Raf-CRD in MAPK signaling, characterized solely in the context of monomeric Ras–Raf-RBD_CRD complexes [24,26,27]. It has been shown in live cells that the CRD is important for the recruitment of Raf to the membrane in the presence of activated Ras [72]. However, the study does not observe insertion of the CRD in the membrane, leaving open other possible mechanisms through which the CRD can have an effect. We also point out that experimental results where the Ras farnesyl group interacts in vitro with the CRD in solution [2,73], are countered by experiments in cells that suggest that the farnesyl group does not contribute to the Ras–Raf-RBD_CRD interaction [74] and recent SPR experiments showing that the farnesyl group does not affect the affinity between KRas and CRaf-RBD_CRD [20]. Thus, it is not clear whether the farnesyl group is involved in the Ras–Raf-RBD_CRD interaction. The SPR experiments that accompany the KRas–CRaf-RBD_CRD structures validate the primary interaction between Ras and the CRD as critical for signaling through MAPK [20], consistent with common interactions in the two alternate models discussed here.

We now know that the surface of the Raf-CRD contains non-overlapping binding sites for Ras (in the monomer of the complex), the Raf kinase domain and the scaffold protein 14-3-3 (Figure 4a). The CRD segments involved in the latter interactions have hydrophobic and positively charged residues that are thought to insert into the membrane once the CRD binds Ras and is released from its autoinhibited state [20,21,24,75,76]. They include residues 143-RKTFLKLAF-151 and 157-KFLLNGFR-164, with the intermitting residues 152-CDICN-156 along with H173 and C176 coordinating the first Zn^2+^-binding site at the N-terminal end of the helix that forms the primary Ras-binding surface of the CRD (Figure 1). These segments are found in the interactions across the Ras dimer interface in our model that provides an alternative role for the CRD in the activated state of Raf. Although these regions are dynamic in NMR structures of the CRD both in solution [3] and in nanodiscs [24], in the crystal structure of the KRas–CRaf-RBD_CRD complex (PDB ID 6XI7), residues 142–146 and 160–164 are part of a β-sheet that presents R143, T145, L160 and N161 for interaction with loop 8 and the ExSAK nucleotide-binding motif of the opposing Ras protomer in the dimer (Appendix A), consistent with our simulations. L160 in particular makes van der Waals contact with the aliphatic portion of Ras K147 of the ExSAK motif across the dimer interface, which in turn interacts with Ras F28 in switch I. Both K147 and F28 are in direct contact with the guanine base of the nucleotide [77]. Several of the Raf-CRD residues in the two segments mentioned above form a core layer of hydrophobic interactions in the CRD that includes packing of L160 at the dimer interface, F151, F158, L159, F146 and the aliphatic portions K144 and R164 (Appendix A). K144 and R164 have the charged portion of their side chains exposed to solvent, capping the end of this hydrophobic cluster opposite from L160. Thus, with the exception of K144 and R164, the polar groups in the supposed membrane-binding segments are arranged at the interface between the CRD and Ras, while the hydrophobic ones form a second layer stabilizing the core structure that presents the Ras-interacting residues. The question of whether the ordering of these key CRD segments is an artifact of our MD simulation model (HRas–CRaf-RBD_CRD) or crystallization (KRas–CRaf-RBD_CRD), or a biologically relevant consequence of dimerization is at the core of the present discussion. We have used the Protein Interfaces Surfaces and Assemblies (PISA) [49] website at the European Bioinformatics Institute to calculate the surface area buried in the KRas–CRaf-RBD_CRD dimer interface (PDB ID 6XI7) as 1005 Å^2^, with a Δ^ι^G *p*-value of 0.496, just within the cutoff indicative of an interaction-specific interface unlikely to be an artifact of crystal packing (see Methods section).

It has been recently demonstrated that mutations of Raf-CRD residues at the primary Ras–Raf-CRD interface only marginally affect the affinity of the Ras–Raf_RBD_CRD complex, dominated by the interaction with Raf-RBD, while these same mutations lead to reduction in the kinase activity in cells [20]. This is consistent with S177A, T182A and M183A variants at the primary Ras-binding site of the CRD inhibiting kinase activity in a previous alanine-scanning mutagenesis study aimed at probing binding epitopes on the surface of the Raf-CRD [74]. Interestingly, point mutations K144A, L160A and R164A also weakly inhibit Raf activation, with the double mutant K144A/L160A completely abrogating signal through Raf [74]. Here, we have two possible interpretations for these results. It can be argued that the inhibitory effects of the K144, L160 and R164 variants are due to disruption of the CRD interaction with the membrane. However, the data are also consistent with disruption of key interactions between the CRD and Ras loop 8 and ExSAK motifs across the dimer interface due to destabilization of the hydrophobic core maintained by key residues in the 143–151 and 157–164 segments of the CRD as described above.

In addition to the inhibitory mutations, the alanine-scanning mutagenesis study identified a number of activating mutations [74], all of which are in residues at or near the binding interface with 14-3-3 and Raf-kinase domain in the autoinhibited state. However, most of these variants, N140A, R143A, T145A, Q156A, K157A, Q166A, T167A, K171A, H175A are activating only in the context of RasG12V, suggesting a Ras dependence on the resulting increased Raf activity. The variants F151A and D153A have significant effects in the context of WT Ras, with a drastic increase in the basal Raf activity that the other mutants do not show [74]. These two residues make key interactions in the inactive state of the kinase [32] and activation is likely due to release of autoinhibition in the absence of Ras. Of the RasG12V dependent activating variants of Raf-CRD, Q156, K157 and Q166, T167 are pairs of hydrophilic residues exposed to solvent that follow one of the cysteine residues in the two loops containing the Zn^2+^-binding pockets. Residue N140 is also exposed to solvent and follows H139 that coordinates one of the Zn^2+^ ions. The mechanism through which mutations of these residues lead to RasG12V-dependent activation of the kinase is currently not known. However, residue K157 is at the beginning of one of the segments thought to interact with the membrane. If this interaction were important for activity, one would expect that the K157A variant would inhibit rather than activate signaling through Raf.

The remaining activating residues in the alanine-scanning mutagenesis study, R143, T145 and H175, make interactions across the dimer interface in the Ras–Raf-RBD_CRD dimer and K171 is part of the packing that stabilizes T145. We have shown that the CRD interactions across the dimer interface are dynamic in the Ras–Raf-RBD_CRD dimer. The activating variants other than F151A and D153A have no effect on WT Ras signaling. However, specific mutations at the dimer interface may synergize with RasG12V to stabilize the dynamic CRD for more efficient information transfer through the allosteric pathways with direct access to the active site and the Ras–Raf-RBD interface. Interestingly, variants associated with RASopathies are also weakly activating [78,79]. In particular, BRaf T241P and Q257R are found in individuals with LEOPARD syndrome and cardiofacialcutaneous (CFC) syndrome, respectively [78]. These residues mediate contacts with the 14-3-3 scaffold dimer in the autoinhibited state. They correspond to T145 and N161 in CRaf, and in the context of the activated Ras–Raf dimer are located at the interface between the CRD and loop 8 across the Ras–Raf-RBD_CRD dimer interface (Figure 2b,c).

The active BRaf/MEK1/14-3-3 structure, with a dephosphorylated 14-3-3-binding motif on the CR2 [32], displays a shift of the 14-3-3 dimer with respect to the Raf-kinase domain relative to the inactive state, accompanied by conformational rearrangement of the Raf-kinase C-terminus, which now redirects away from the Ras-binding surface of the Raf-CRD (Figure 4b). The absence of the Raf-RBD from the inactive BRaf cryo-EM reconstruction suggests it does not contribute toward stabilizing the autoinhibited state and therefore must remain accessible for Ras-binding [32]. This is consistent with a highly dynamic RBD observed by MD simulations [80]. Indeed, by aligning the CRaf-CRD from our Ras–Raf-RBD_CRD structure with the BRaf-CRD in the inactive complex, we observe that the short Raf-RBD_CRD linker is sufficient for the Raf-RBD to be accessible for interaction with Ras (Figure 4c and Appendix A). We propose that Ras-binding to the Raf-RBD results in the dissociation of the 14-3-3 subunit from the CR2-binding site and redirection of the Raf C-terminus. This would expose the CR2 site for dephosphorylation, preventing association of the 14-3-3 dimer to reestablish Raf autoinhibition. This is consistent with an increase in Raf-kinase activity in cells upon dephosphorylation of the CR2 site [5,29,33]. The displacement of 14-3-3 from the CR2-binding site would promote its interaction with a second Raf kinase to facilitate kinase domain dimerization and further expose the Raf-CRD for extraction (Figure 4b,c), resulting in collapse of the autoinhibited conformation and progression toward Raf activation.

Our proposed model for Ras-mediated activation of the MAPK pathway (Figure 4c) is similar to the recently published model [20] in the mechanism by which Raf-RBD binds to Ras. However, the two models deviate in that, in our model, binding of Raf-RBD to Ras promotes Ras dimerization, permitting allosteric modulation of the Galectin-binding regions on the Raf-RBD, to form a platform of Ras–Raf and Galectin dimers for signal amplification (Figure 3e). During this process the CRD is released from its autoinhibitory role, allowing the shift in 14-3-3 and stabilization of the Raf dimer. Given the high-affinity dimerization of Ras promoted by Raf-RBD that we previously reported [8], we propose that these events occur concertedly, resulting in simultaneous release of autoinhibition and allosteric modulation of the Galectin-binding site on Raf-RBD, with robust activation of the Ras–Raf–MEK–ERK pathway.

Our model of activation of Raf through formation of the Ras dimer upon binding Raf-RBD serves as a viable alternative to the recently published model of a monomeric Ras–Raf-RBD_CRD complex in which the CRD is inserted in the membrane [20]. Both models remain to be tested and validated, as there is currently not enough experimental evidence to prove either one. Although not discussed here, a third model of Ras activation of Raf was published as a preprint in BioRXiv [81] and that model too remains to be tested. We, in the Ras research community, have made exciting progress toward a better understanding of the Ras–Raf interaction and of the components important in the release of autoinhibition and activation of the Raf kinase domain. We must now stay open-minded as we consider the mechanistic link that lies between the binding of Raf-RBD to Ras and the resulting robust signaling output in cells. Given the artificial contexts in which the CRD has been observed to bind membranes, and the fact that there are examples in the literature where artificial protein–membrane interactions have been observed [82,83,84], the question of whether the CRD contributes to membrane insertion or to interaction across the Ras–Raf-RBD_CRD dimer interface remains open and must be further explored. Our recent demonstration that Raf-RBD promotes dimerization of Ras on the membrane and that this results in strong allosteric connections across the dimer formed through α4 and α5, provides an important set of constraints in going forward with refining the model for Ras activation of Raf. The placement of the CRD in this dimer at a strategic position to affect allosteric communication between functionally important regions of Ras, and beyond to the Galectin-binding regions of Raf-RBD, calls for serious consideration as we move forward.

## 5. Conclusions

The crystal structure of the HRas–CRaf-RBD_CRD complex presented here with our analysis of the dimer model based on MD simulations supported by a recently published structure of the KRas–CRaf-RBD_CRD in which the dimer is present provide a key missing link in our understanding of Ras-mediated Raf activation. They show a continuous surface for Ras–Raf interaction that links the Ras active site and dimerization interface through switch I, loop 3, and α5. In the context of the dimer, the CRD also contacts loop 8 and residues in two highly conserved nucleotide-binding motifs, centrally positioned to access regulatory and functional regions of Ras. This is consistent with the observation that the Raf-CRD is required for Raf activation in vivo. Our analysis of these structures in the context of other recent structural breakthroughs in the literature provides an emerging model of Ras activation of Raf, where formation of the Ras–Raf complex is linked to both dimerization and release of Raf autoinhibition to form a platform for amplified synchronized signaling. This model builds upon our recent proposal of higher-order Ras–Raf–Galectin signaling assemblies, paving forward a new research direction paramount to our understanding of signaling through the Ras–Raf–MEK–ERK pathway [8]. Because the affinity between Ras and the Raf-CRD alone is relatively weak, the disruption of its interactions with Ras, either through the interswitch region involving loop 3 or through loop 8, holds strong potential as a novel approach for targeting Ras- and Raf-related cancers. The structural analysis presented here, with release of the requirement for insertion of the CRD in the membrane, provides an alternate framework, consistent with available data on Ras dimerization and nanoclustering, from which to explore the effects of oncogenic mutants on signaling through the MAPK pathway.

## Figures and Tables

**Figure 1 biomolecules-11-00996-f001:**
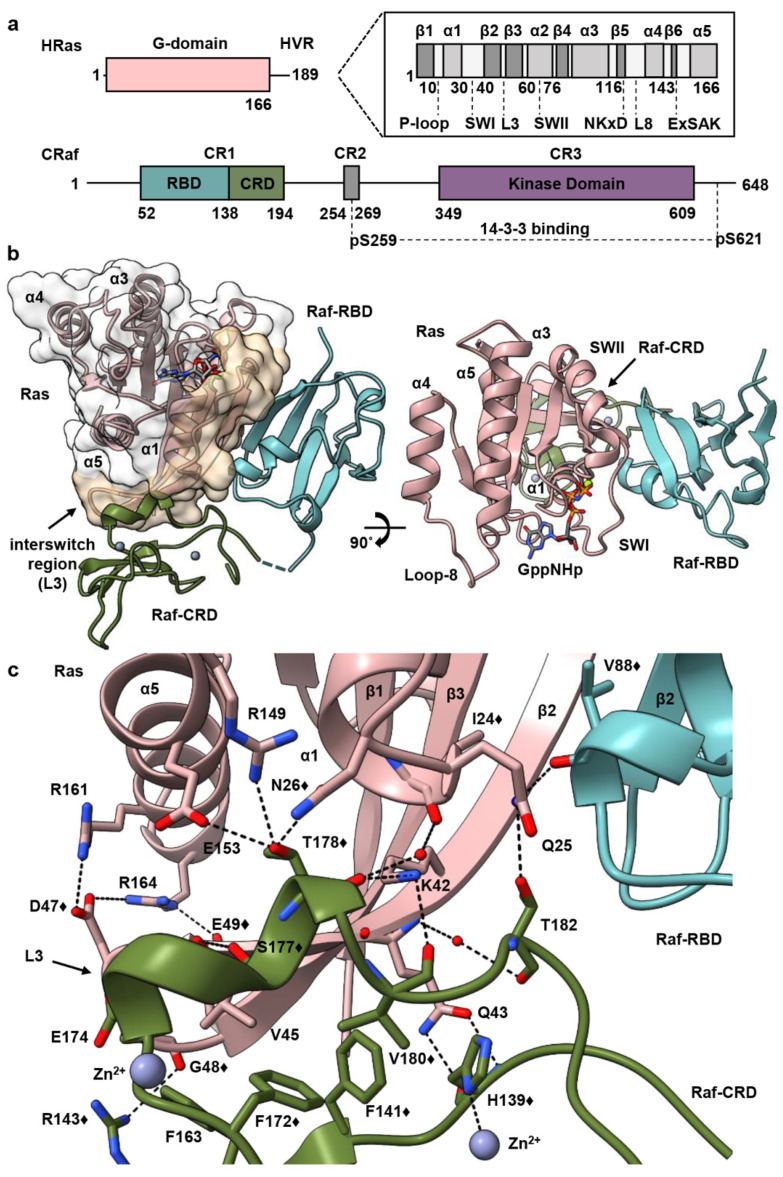
Structure of the HRas–CRaf-RBD_CRD complex. (**a**) Schematic diagrams of HRas and CRaf provide structural boundaries for the Ras G domain and hypervariable region, and the three conserved regions found in Raf kinases. (**b**) Cartoon and surface representations of the HRas–CRaf-RBD_CRD complex reveal a continuous binding surface spanning Ras (pink), residues 23 through 48 for both the Raf-RBD (teal), and Raf-CRD (green). (**c**) The Raf-CRD interacts with residues in Ras α1, α5 and the interswitch region through hydrogen bonds and hydrophobic contacts. The symbol ♦ identifies residues that show chemical shift perturbations in the analysis described by Feng et al. [24].

**Figure 2 biomolecules-11-00996-f002:**
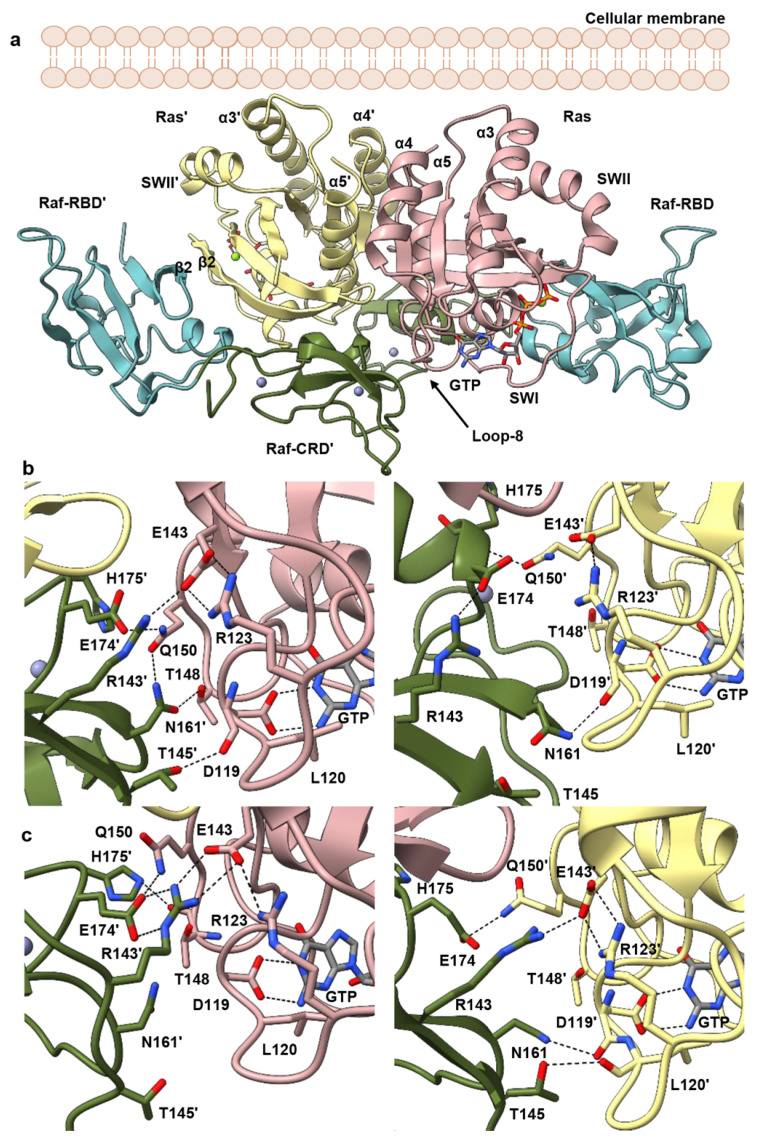
HRas–CRaf-RBD_CRD dimer complex. (**a**) Modelled HRas–CRaf-RBD_CRD dimer complex. Approximately perpendicular orientation of helices 3, 4 and 5 with respect to the membrane does not allow the Raf-CRD to interact with the membrane. (**b**) Frame taken from the replicate 2 HRas–CRaf-RBD_CRD simulation after 60 ns shows sample Ras–Raf-CRD’ (**left**) and Ras’/Raf-CRD (right) interactions. (**c**) Frame taken from replicate 3 HRas–CRaf-RBD_CRD simulation after 140 ns shows sample Ras–Raf-CRD’ (**left**) and Ras’/Raf-CRD (**right**) interactions. Right and left panels in b and c illustrate the asymmetry introduced through the Raf-CRDs’ dynamic interaction with the opposing Ras protomers. One protomer of HRas in the dimer is shown in pink and the other, HRas’ in yellow. Both CRDs in the dimer, CRD and CRD’ are shown in dark green.

**Figure 3 biomolecules-11-00996-f003:**
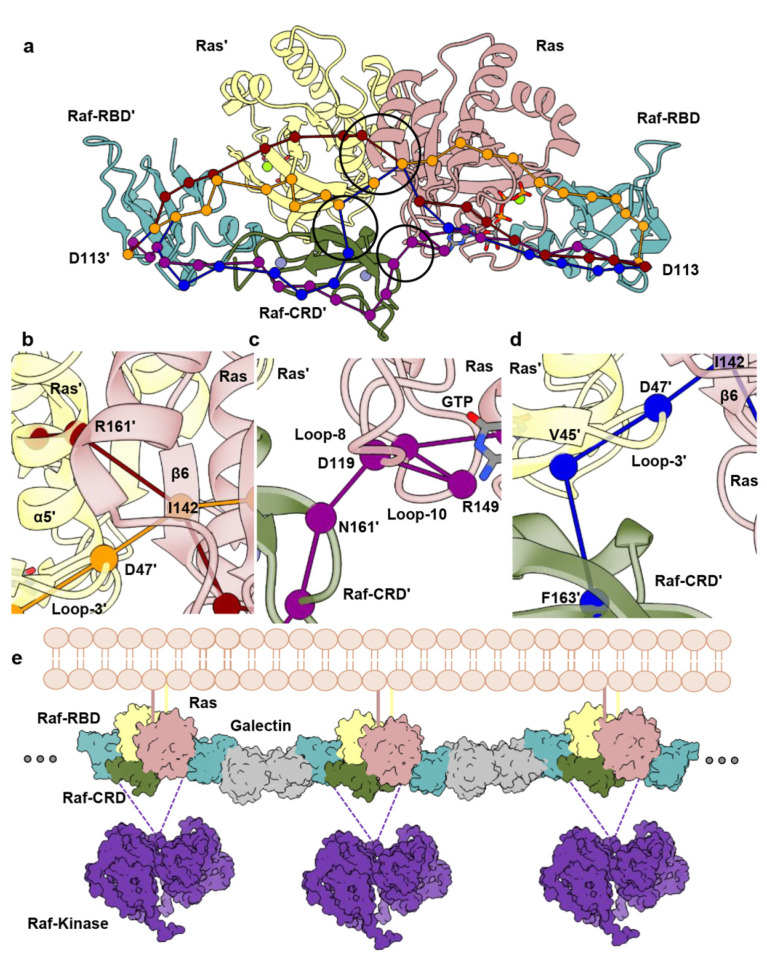
Dynamical network analysis performed for HRas–CRaf-RBD_CRD dimer simulations. (**a**) Allosteric paths calculated between D113 and D113’ of the Raf-RBD molecules. (**b**) Close-up view showing path for intermolecular information involving Ras loop 3/β6 (orange) and Ras α5/β6 (red) across the Ras dimerization interface. (**c**) Close-up view showing path for intermolecular information involving Raf-CRD/Ras loop 8 (purple). (**d**) Close-up view showing path for intermolecular information involving Raf-CRD/Ras interswitch region (blue). (**e**) Proposed Ras–Raf–Galectin assemblies adapted from Packer et al. [8] with inclusion of the Raf-CRD. Galectin (grey) (PDB ID 3W58) and Raf-kinase (purple) (PDB ID 6Q0J) were downloaded from the Protein Data Bank. The scaffold protein 14-3-3 stabilizes the Raf kinase dimer (not shown). Given that the structure of the CR2 is unknown, it is also possible that kinase domains from distinct Ras–Raf-RBD_CRD dimers bridged by Galetin come together to form the activated Raf dimer.

**Figure 4 biomolecules-11-00996-f004:**
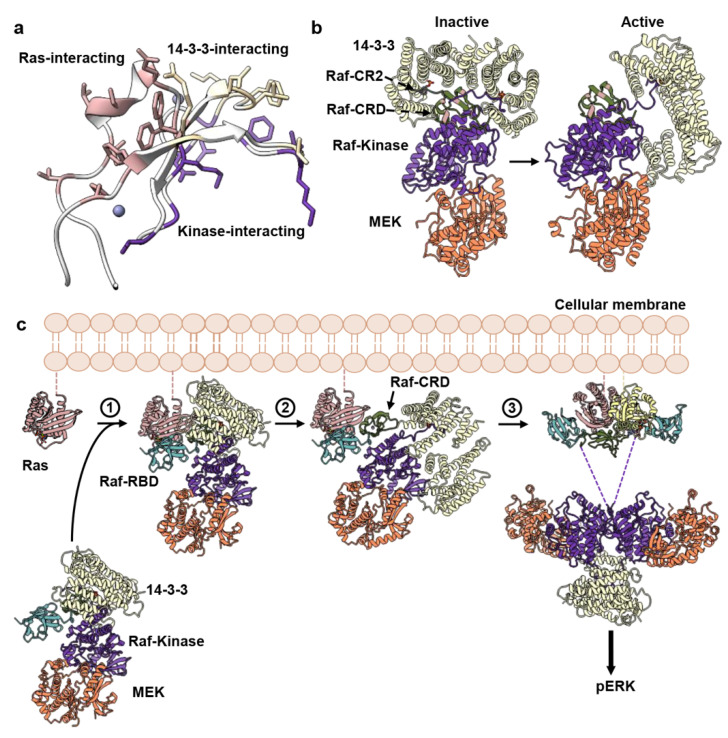
Proposed Mechanism for Ras-mediated activation of Raf. (**a**) Residues involved in the primary Ras–Raf-CRD interaction (pink) do not overlap with Raf-CRD residues used to stabilize contacts with the kinase domain (purple) and 14-3-3 proteins (wheat) in the autoinhibited state. (**b**) Comparison of Raf-kinase domain (purple) and 14-3-3 (wheat) conformations in the inactive (PDB ID 6NYB) and active Raf structure (PDB ID 6Q0J). Note the shift in the 14-3-3 dimer in the absence of the Raf CR2 motif and release of the Raf-CRD. (**c**) Proposed structural model for the Ras-mediated release of Raf autoinhibition and activation of MAPK signaling. Raf-RBD (cyan) is exposed in the autoinhibited state, disordered in the cryo-EM structure, likely due to flexibility in its linker to the CRD (dark green). The high-affinity binding of Raf-RBD to Ras places the exposed area of the CRD in its Ras-binding site, while steric hindrance is expected to displace the Raf C-terminus (purple) and promote the shift of the 14-3-3 dimer (wheat) towards its active state interaction with a second Raf kinase domain (purple). These two features of the Ras–Raf interaction are shown in steps 1 and 2. Binding of Raf-RBD concurrently promotes dimerization of the complex as shown in step 3, which we propose is a key step in binding Galectin dimers to build a platform for signal amplification (see Figure 3e). We propose that steps 1, 2 and 3 occur concertedly to activate Raf.

## Data Availability

Crystallographic coordinates and structure factors were deposited in the Protein Data Band with accession code 7JHP.

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
