# Peer review of "Crystal Structure Reveals the Full Ras–Raf Interface and Advances Mechanistic Understanding of Raf Activation"

_biomolecules, 2021, doi:10.3390/biom11070996_

Round 1

Reviewer 1 Report

The authors present the analysis of the HRas/CRaf-RBD_CRD complex using a combination of methods, especially X-ray crystallography and MD simulations. The topic is timely and the results, well presented. The discussion is also well structured and provides detailed explanations of the mechanism leading to Raf activation. The possibility of dimerisation of the Ras/Raf complex is highlighted to promote signal amplification. This is fairly interesting. My point is in the involvement of Galectins in the process. Looking at the references, it seems that it is Galectin-1 the lectin involved in the interaction with the dimer and not any other Galectin. Indeed, galectin-1 is a homodimer (it also displays allosterism upon glycan binding: Chem. Eur. J. 2020 Dec 1;26(67):15643-15653), but not Galectin-3, which is chimera-type Galectin. Therefore, the authors should explicitly recognise galectin-1 as the target and propose an interaction model. Which is the region of Galectin-1 involved in binding to the complex? Is there any relationship with the glycan binding site at the galectin? Can the supramolecular complex formation be modulated by other glycans that interfere with Galectin interactions? Otherwise, the manuscript shows a high quality and merits publication after these points are addressed.

Author Response

Galectin-1 and Galectin-3 function as scaffold proteins in HRas and KRas signaling respectively. It was originally thought that Galectins bound directly to Ras, but Blazevits et al (ref 13) showed that Galectin-1 binds directly to Raf-RBD, not HRas, and proposed a computational model of the complex using molecular docking validated by mutational experiments. There are no experimental structures of the Raf-RBD/Galectin-1 complex, but in the computational model the Raf-RBD binding site on Galectin-1 partially overlaps with its glycan binding pocket. I was excited to read the paper, referenced by the reviewer, showing that binding of oligosaccharides, particularly LacNAc to Galectin-1 promotes allosteric connections to the dimer interface. It is possible that binding of Raf-RBD could also promote this effect, allosterically connecting the signaling platform that we propose for the Ras/Raf and Galectin-1 dimers (Figure 3 in text). We add a brief suggestion of this in the discussion section, including the reference from Chem. Eur. J. (2020). While Galectin-1 is always found as a homodimer, Galectin-3 is a monomer when bound to glycans but forms dimers in its apo form in the absence of lactose or other sugars (Yang et al, Biochemistry, 1998). We now mention in the discussion that we would expect Galectin-3 to play a role in KRas signaling analogous to the one we propose for Galectin-1 in signaling through HRas.

Reviewer 2 Report

The manuscript by T. Cookis and C. Mattos reports the crystal structure of the G-domain of HRas (aa 1-66) in complex with CRaf-RBD-CRD (aa 52-187). The work also reports molecular dynamics simulations performed on the constructed dimer in three runs of 350 ns each including node trajectories analysis. As the protein complex crystallizes with one copy in the asymmetric unit, the authors generate the Ras/Raf dimer by 2-fold crystallographic symmetry as observed in the crystal structure of the HRas/RafRBD (PDB ID 4G0N) the Mattos group has previously reported.

The work is significant to the Ras-field as questions regarding how Ras forms nanoclusters on cell membranes, what Ras surfaces are involved, and if Raf (and/or other effectors) drives the formation of higher oligomeric complexes are still pending. The interface between the two proteins is similar to the recently published KRas/Raf complex (ref 16, Tran et al., Nat Commun 2021). The main difference between the two reports is in the interpretation of the results and implication for signaling. Here, the authors argue that Ras dimers are the signaling unit that provides a platform for Raf dimerization and signal amplification while in ref 16, Ras is not required to dimerize to activate Raf kinase. The work is a continuation of a recent PNAS paper (Packer et al. 2021) Dr. Mattos published in collaboration with the Groves group showing that RafRBD promotes Ras dimerization on model membranes.

Overall, the experimental procedures are well described. The major weakness of the manuscript is in the conclusion, which touches on every aspect of Ras signaling, including Raf activation and the role of 14-3-3 proteins yet the only experimental data presented is the crystal structure of two truncated proteins lacking major domains. The center piece of authors’ argument is that Ras dimerizes, and that the dimers are the basic unit for the observed nanoclusters in cells. The dimerization of Ras is a theme that is repeated over and over in the manuscript even though the authors had to generate the dimers from two-fold axis of symmetry. Few things the authors can report and do to further validate their model and arguments:

  • First, what is the accessible surface area buried at the Ras dimer, what is the percentage of hydrophilic and hydrophobic buried surface areas, and how does the buried interface compare to other protein-protein interfaces with similar affinity?
  • Second, a model remains a model until it is validated experimentally. The authors should disrupt the interface or promote dimer formation by site-directed mutagenesis and study effects of generated mutants on Ras signaling.
  • Third, in the proposed model, the CRD is at the base of the Ras dimer interface allosterically connecting key regions on Ras involved in nucleotide binding and stabilization. But what to make of other Ras effectors and Ras-binding proteins that contain the Ras Association (RA) domain equivalent to RafRBD but not a CRD or equivalent? Would Ras in this context also signal as a dimer or is Ras dimerization specific to Raf?
  • The MD simulations are interesting but are contingent on the validity of the model (see my 2nd point). MD simulation results should be validated experimentally by site-directed mutagenesis. For example, if the Ras E143 – Raf R143 interaction is judged important, what disrupting it would do to the complex.
  • Figure S3b: in replicate 1 through 3, what are cluster1, 2, and 3 and how are they defined? These clusters are not described or referred to in the text.

Other points to be addressed:

  • Galectin and its role in modulating the Ras/MAPK should be properly introduced in the Introduction section to the general audience.
  • Line 46: nanoclustering instead of nonoclustering.
  • Line265: ‘Various of the interactions revealed..’ sentence does not read right. Is it ‘Various interactions..’?
  • Figure 1C: legend says Ras helix 1 but figure shows 5. Also, no need to label water molecules as this panel is already busy and since waters are not referred to in the text.
  • Figure 1A shows Ras secondary structures using Greek letters but the text uses helix and strand followed by a number. The text and figures should be consistent.
  • Why is the HRasR97C called allosteric mutant? What is allosteric about it?
  • If CRD L160 is a critical residue in stabilizing (stabilizes how, what is meant by stabilize?) the nucleotide binding site why not mutate it and test implication on GTP-release. In this regard, Hermann et al. JBC 1995 showed that RBD alone is sufficient to inhibit nucleotide release, so what is L160 and/or the hydrophobic cluster adding.
  • The Discussion section is too long and should relate to the data presented in the manuscript. Are the two paragraphs ‘The cryo-EM structure… progression toward Raf activation’ (lines 498-525) in the Discussion section necessary as they are not pertinent to the data presented in the manuscript?

Author Response

The manuscript by T. Cookis and C. Mattos reports the crystal structure of the G-domain of HRas (aa 1-66) in complex with CRaf-RBD-CRD (aa 52-187). The work also reports molecular dynamics simulations performed on the constructed dimer in three runs of 350 ns each including node trajectories analysis. As the protein complex crystallizes with one copy in the asymmetric unit, the authors generate the Ras/Raf dimer by 2-fold crystallographic symmetry as observed in the crystal structure of the HRas/RafRBD (PDB ID 4G0N) the Mattos group has previously reported.

The work is significant to the Ras-field as questions regarding how Ras forms nanoclusters on cell membranes, what Ras surfaces are involved, and if Raf (and/or other effectors) drives the formation of higher oligomeric complexes are still pending. The interface between the two proteins is similar to the recently published KRas/Raf complex (ref 16, Tran et al., Nat Commun 2021). The main difference between the two reports is in the interpretation of the results and implication for signaling. Here, the authors argue that Ras dimers are the signaling unit that provides a platform for Raf dimerization and signal amplification while in ref 16, Ras is not required to dimerize to activate Raf kinase. The work is a continuation of a recent PNAS paper (Packer et al. 2021) Dr. Mattos published in collaboration with the Groves group showing that RafRBD promotes Ras dimerization on model membranes.

Overall, the experimental procedures are well described.

The major weakness of the manuscript is in the conclusion, which touches on every aspect of Ras signaling, including Raf activation and the role of 14-3-3 proteins yet the only experimental data presented is the crystal structure of two truncated proteins lacking major domains.

The model we propose for Ras/Raf dimers has direct implications for Raf activation in the presence of 14-3-3 proteins. The point here is not to prove any of the models but to show a viable alternative based on our model MD simulations that is consistent with available data, if we allow for the possibility that CRD membrane insertion is an artifact. We would also like to point out that our model, although generated from our structure of the HRas/Raf-RBD dimer, is validated by the existence of this same dimer in the KRas/Raf-RBD_CRD crystal structure published by Tran et al, but not noted by them in their paper. One of the purposes of the discussion is to place the results in the larger context of what is known in the literature. Tran et al did exactly this when using their model, obtained from similarly truncated proteins, to propose how Ras activates Raf, with the same 14-3-3 and kinase domains as we do here. In my opinion that is entirely appropriate for them as it is for us to do. Here we draw on the known structures of Raf/14-3-3/MEK complexes available in the literature to complement our structures in the development of an alternative model of Ras activation of Raf than the current model prevalent in the literature based on the assumption that the CRD must be inserted in the membrane. Furthermore, we draw on the vast literature of experimental mutational analysis in vitro and in cells to show that mutational analysis involving the CRD in cells does not resolve between the two models regarding interactions across the dimer interface.

The center piece of authors’ argument is that Ras dimerizes, and that the dimers are the basic unit for the observed nanoclusters in cells. The dimerization of Ras is a theme that is repeated over and over in the manuscript even though the authors had to generate the dimers from two-fold axis of symmetry.

Response: It is correct that the center piece of this paper is our proposal that Ras dimerizes when signaling through Raf. Thus, our frequent mention of Ras dimers. Dimerization through the a4-a5 interface places the CRD near functionally relevant regions of Ras both in the monomer as previously described and across the dimer interface as we point out here for the first time. With this placement of the CRD we point to the possibility that the CRD insertion in the membrane is not a biologically relevant phenomenon. This is a novel contribution that deviates from a deeply rooted notion that currently prevails without unequivocal proof that it occurs in cells. The dimer exists in the crystals of HRas/CRaf-RBD and of KRas/CRaf-RBD_CRD, each in a different crystal form. We now specify on line 105 that the KRas/CRaf-RBD_CRD dimer is present in the crystal, generated by a two-fold crystallographic symmetry axis from the asymmetric unity containing the complex monomer (PDB ID 6XI7). In the case of our model of the HRas/CRaf-RBD_CRD dimer, it was not present in the crystal, such that we had to model it from our crystal structure of the HRas/CRaf-RBD dimer that was. Our resulting model is very similar to that in the KRas/CRaf-RBD_CRD crystal structure with various of the experimentally observed interactions sampled in our simulations as described in the results section. We are not claiming that our model is correct. However, we do contend that it is as viable as the currently prevalent model and should be considered as an alternative until the existing models are thoroughly tested.

Few things the authors can report and do to further validate their model and arguments:

  • First, what is the accessible surface area buried at the Ras dimer, what is the percentage of hydrophilic and hydrophobic buried surface areas, and how does the buried interface compare to other protein-protein interfaces with similar affinity?

Response: We have used the PISA website to calculate the buried surface area at the KRas/Raf-RBD_CRD dimer interface found in the structure with PDB ID 6XI7 to be 1005 Å2. We also determined that the hydrophobic content of the interface is indicative of an interaction-specific interface, unlikely to be an artifact of crystal packing. We included a brief description of PISA in the Methods with reference to the website and added lines 522 - 530 in the Discussion regarding these results.

  • Second, a model remains a model until it is validated experimentally. The authors should disrupt the interface or promote dimer formation by site-directed mutagenesis and study effects of generated mutants on Ras signaling.

Response: Data for mutational analysis of the CRD at residues interacting across the dimer interface, with focus on Ras signaling outcomes, are already available in the literature and discussed in the paper. There are both activating and deactivating mutations and we analyze each one in the context of our model. One of the problems with interpreting the results in terms of the available models is that the site that interacts across the dimer interface overlaps with the interactions of the CRD in the auto-inhibited state of the kinase in the presence of 14-3-3. Thus, the causal effects of the signaling outcomes are ambiguous in terms of the available models.

  • Third, in the proposed model, the CRD is at the base of the Ras dimer interface allosterically connecting key regions on Ras involved in nucleotide binding and stabilization. But what to make of other Ras effectors and Ras-binding proteins that contain the Ras Association (RA) domain equivalent to RafRBD but not a CRD or equivalent? Would Ras in this context also signal as a dimer or is Ras dimerization specific to Raf?

Response: In our opinion this question is beyond the scope of the present paper. The literature has reported dimerization in cells for signaling through MAPK and we have shown that Ras-RBD promotes dimerization of Ras. To our knowledge, the question of whether other RBDs promote dimerization of Ras has not yet been explored either in cells or in vitro and thus must remain open. For the present paper we limit our discussion of dimerization to the Ras/Raf interaction.

  • The MD simulations are interesting but are contingent on the validity of the model (see my 2nd point). MD simulation results should be validated experimentally by site-directed mutagenesis. For example, if the Ras E143 – Raf R143 interaction is judged important, what disrupting it would do to the complex.

Response: As mentioned in our response to the reviewer’s 2nd point, the results of mutational analysis of residues that interact across the dimer interface are not easy to interpret in terms of the model. In 1998 Jonathan Cooper’s lab, in collaboration with Deborah Morrison and Sharon Campbell published a paper where they used a yeast two-hybrid system to select for mutants of Raf that control Ras binding and activity in cells (Winkler et al, JBC, 1998, vol 34, pgs 21578-21584). CRD residue R143 mutations showed up as R143W and R143Q. These mutants activated Raf signaling and the authors proposed that this was due to decrease in inhibitory interactions between the N-terminal regulatory regions and the kinase domain. They were also shown to bind more strongly to Ras using enzyme-linked immunosorbent assays. Today we know that CRD R143 makes key interactions with 14-3-3 in the auto-inhibited state and this could explain the activating effects of mutations at these residues. In our model of the dimer, a tryptophan at CRD residue 143 would stack against CRD F163 and make good van der Waals interaction with the aliphatic portion of Ras R123 across the dimer interface, possibly explaining tighter binding to Ras. In the CRD R143Q mutant the glutamine side chain would make good H-bonding interactions with both R123 and E143 in the Ras salt bridge, again across the dimer interface. Thus, it does not appear that the mutants would have a deleterious effect on the dimer, although it would be difficult to test this in solution due to the small amount of dimer present even in the presence of Raf (ref 8). Meaningful mutational analysis to test biochemical function in vitro will be complex and beyond what we aim to do here. For the present we are suggesting an alternative model consistent with available data which we and others can test in the future.

  • Figure S3b: in replicate 1 through 3, what are cluster1, 2, and 3 and how are they defined? These clusters are not described or referred to in the text.

Response: The clusters are associated with different values of RMSD. They do not add much to our analysis and as the reviewer pointed out we don’t reference it in the text. We therefore have removed Figure S3b and renamed S3c to S3b in the figure and text.

Other points to be addressed:

  • Galectin and its role in modulating the Ras/MAPK should be properly introduced in the Introduction section to the general audience.

Response: We added a few sentences starting on line 47 to introduce Galectins.

  • Line 46: nanoclustering instead of nonoclustering.

Response: Corrected, thank you.

  • Line265: ‘Various of the interactions revealed..’ sentence does not read right. Is it ‘Various interactions..’?

Response: We changed to “Various interactions…” as proposed by the reviewer.

  • Figure 1C: legend says Ras helix 1 but figure shows 5. Also, no need to label water molecules as this panel is already busy and since waters are not referred to in the text.

Response: Actually, it is both. We changed the legend to a1 and a5 and changed the a1 label position. We also deleted the water molecule labels as suggested.

  • Figure 1A shows Ras secondary structures using Greek letters but the text uses helix and strand followed by a number. The text and figures should be consistent.

Response: We changed all text references of secondary structure to Greek letters for consistency with the figures.

  • Why is the HRasR97C called allosteric mutant? What is allosteric about it?

Response: This language is internal to our laboratory, referring to experiments that are currently ongoing. It should not have been included. We have removed mention of allosteric site and now simply state that we used the HRasR97C mutant to obtain our structure of the complex.

  • If CRD L160 is a critical residue in stabilizing (stabilizes how, what is meant by stabilize?) the nucleotide binding site why not mutate it and test implication on GTP-release. In this regard, Hermann et al. JBC 1995 showed that RBD alone is sufficient to inhibit nucleotide release, so what is L160 and/or the hydrophobic cluster adding.

Response: The problem here again is obtaining sufficient amount of the Ras/Raf-RBD_CRD dimer to do the suggested experiments in solution, as the predominant species is the monomer even at high concentrations in the absence of the membrane (ref 8). We changed the language in the text to simply state the interaction between Raf L160 and Ras K147, rather than suggest that it stabilizes the active site.

  • The Discussion section is too long and should relate to the data presented in the manuscript. Are the two paragraphs ‘The cryo-EM structure… progression toward Raf activation’ (lines 498-525) in the Discussion section necessary as they are not pertinent to the data presented in the manuscript?

Response: We have shortened the two paragraphs to provide the minimum information necessary to understand our proposed model of Ras activation of Raf.

Reviewer 3 Report

This study is of high quality and interest to the scientific community and the results presented here are valuable in so far as it may be necessary to revise the current opinion about the Raf activation mechanism. The authors have a substantial publication history in this particular topic and the current work is a continuation of their quality research on Raf/Ras kinase interactions. The novel results are presented in a very detailed way including multiple illustrative schemes, which are crucial for understanding of the complex conformation, especially with respect to the CRD domain. 

In the results section, a plethora of details concerning the orientation of the individual domains, interactions between specific residues are given. This section is rather difficult to read for a person who is not familiar with the Ras/Raf system, however informative it may be. Some stylistic changes (shortening of some long sentences) might improve the overall “digestibility”. 

The authors mention structural differences between their crystallographic structure of Raf-CRD and the previously published NMR structure (Mott et al., 1996). This particular region has been described as highly dynamic also in the other NMR structure (Fang et al., 2020) where this domain adjoins the lipid bilayer. Did the authors consider the possibility that the beta sheet present in their structure might be a crystallization/MD-driven artefact? This issue should be discussed.

One minor mistake - in the Supplementary material, Fig. S2 is wrongly annotated (“c” section in the picture is labeled as “a”).

Author Response

This study is of high quality and interest to the scientific community and the results presented here are valuable in so far as it may be necessary to revise the current opinion about the Raf activation mechanism. The authors have a substantial publication history in this particular topic and the current work is a continuation of their quality research on Raf/Ras kinase interactions. The novel results are presented in a very detailed way including multiple illustrative schemes, which are crucial for understanding of the complex conformation, especially with respect to the CRD domain. 

Response: Thank you for the complementary comment.

In the results section, a plethora of details concerning the orientation of the individual domains, interactions between specific residues are given. This section is rather difficult to read for a person who is not familiar with the Ras/Raf system, however informative it may be. Some stylistic changes (shortening of some long sentences) might improve the overall “digestibility”. 

Response: Point taken. We have shortened a few sentences and rewritten others for better readability/digestibility. Although a bit heavy for readers non-experts in the Ras/Raf system, our descriptions provide information that we later use in the discussion in the context of mutational analysis results from the literature.

The authors mention structural differences between their crystallographic structure of Raf-CRD and the previously published NMR structure (Mott et al., 1996). This particular region has been described as highly dynamic also in the other NMR structure (Fang et al., 2020) where this domain adjoins the lipid bilayer. Did the authors consider the possibility that the beta sheet present in their structure might be a crystallization/MD-driven artefact? This issue should be discussed.

Response: The question of whether the ordering of the CRD loops in particular and dimerization in general is an artifact of crystallization (in case our model is not correct) or a consequence of the biologically relevant dimer (consistent with our model) is at the core of the present paper. The possibility of an artifact of CRD membrane insertion in vitro is key to our alternative model and still needs to be tested, as does our proposed biological role for dimerization. If the dimer is an artifact it would be an artifact that appears repeatedly in several crystal structures, including Ras, Ras/Raf-RBD and Ras/Raf-RBD_CRD.  We have added the fact that the CRD loops are also flexible in the nanodisc embedded NMR structure (Feng et al) and at the end of that same paragraph explicitly point out the possible artifact of ordering of the b-sheet being at the core of the discussion. Furthermore we use the PISA website at the EBI to calculate the dimer interface area and determine the DG P-value, which is indicative that the interface is unlikely to be an artifact of crystallization (page 12, lines 522-530).

One minor mistake - in the Supplementary material, Fig. S2 is wrongly annotated (“c” section in the picture is labeled as “a”).

Response: This is now corrected, thank you.

Reviewer 4 Report

General comment:

This manuscript, entitled “Crystal structure reveals the full Ras:Raf interface and advances mechanistic understanding of Raf activation,” authored by Cookis and Mattos reports the 2.8 Å crystal structure of the HRas/CRaf-RBD_CRD complex showing the Ras/Raf interface as a continuous surface on Ras, similar to the KRas/CRaf-RBD_CRD structure. Based on modeled dimer and MD simulation, they proposed a molecular model in which Ras binding is involved in the release of Raf autoinhibition. At the same time, the Ras/Raf complex dimerizes to promote a platform for signal amplification, with Raf-CRD centrally located to impact regulation and function. This kind of approach is suitable for connecting the structure-function relationship of the interacting protein molecules to understand the mechanistic role in the biological system. In my opinion, this is a valuable work and is suitable for publication in biomolecules after the authors have addressed the following comments and questions:

Specific comments:

  • Is there any structural/conformational information available for inactivated Raf-kinase in the absence of Ras?
  • How is autoinhibition of Raf-kinase regulated?
  • What is the functional relevance of dimerization of HRAS:RBD–CRD complex near the membrane, other than allostery? Is there any experimental evidence of allostery that enhances phosphorylation and signaling pathways in this case?
  • The structural contact between CRD and RAS suggest hydrophobically– but there are enough electrostatic or salt bridge interaction – Gln66-Arg41, Ser39- Arg89/Arg67, Glu37-Arg59 – please elaborate these interactions.

Author Response

This manuscript, entitled “Crystal structure reveals the full Ras:Raf interface and advances mechanistic understanding of Raf activation,” authored by Cookis and Mattos reports the 2.8 Å crystal structure of the HRas/CRaf-RBD_CRD complex showing the Ras/Raf interface as a continuous surface on Ras, similar to the KRas/CRaf-RBD_CRD structure. Based on modeled dimer and MD simulation, they proposed a molecular model in which Ras binding is involved in the release of Raf autoinhibition. At the same time, the Ras/Raf complex dimerizes to promote a platform for signal amplification, with Raf-CRD centrally located to impact regulation and function. This kind of approach is suitable for connecting the structure-function relationship of the interacting protein molecules to understand the mechanistic role in the biological system. In my opinion, this is a valuable work and is suitable for publication in biomolecules after the authors have addressed the following comments and questions:

Specific comments:

  • Is there any structural/conformational information available for inactivated Raf-kinase in the absence of Ras?

Response: Yes. A series of cryo-EM structures were published simultaneously last year for B-Raf in various states by the Kuriyan and Eck laboratories (refs 32 and 38). The inactive Raf kinase structure in the absence of Ras is published with PDB ID 6NYB and shown in Figure 4b.

  • How is autoinhibition of Raf-kinase regulated?

Response: This question is complex and somewhat beyond the scope of the present article. We have indicated some regulatory components most relevant to our model, such as Ras-RBD binding and dephosphorylation of a CR2 residue involved in 14-3-3 interaction. There were originally two paragraphs in the discussion describing this process. We have significantly shortened this as requested by reviewer 2, leaving only the bare minimum relevant to the description of our model of Ras activation of Raf.

  • What is the functional relevance of dimerization of HRAS:RBD–CRD complex near the membrane, other than allostery? Is there any experimental evidence of allostery that enhances phosphorylation and signaling pathways in this case?

Response: The functional relevance of dimerization of the HRas/Raf-RBD_CRD complex is still under debate in the literature. In our model, the relevance of Ras dimerization is the coupling to Galectin dimer through allosteric modulation of the Galectin binding site on Raf-RBD to form a signaling platform for signal amplification. This is explicitly mentioned in the second paragraph of the discussion. The connection between allostery and phosphorylation resulting from active signaling has not yet been made experimentally.

  • The structural contact between CRD and RAS suggest hydrophobically– but there are enough electrostatic or salt bridge interaction – Gln66-Arg41, Ser39- Arg89/Arg67, Glu37-Arg59 – please elaborate these interactions.

Response: The electrostatic interactions pointed out by the reviewer are between RBD residues and Ras. This interface is indeed highly electrostatic, populated by several salt bridges, and has been described elsewhere (ref 7 in main text, page 4).

Round 2

Reviewer 2 Report

In their revised manuscript, the authors addressed some of the points I raised in my previous critique. Unfortunately, they only addressed points related to the presentation of the manuscript. The main issue related to the validity of their model was not addressed. For example, not a single mutant at the Ras dimer interface or to validate the molecular dynamics results was generated as requested. Repeated arguments devoid of experimental data to backup claims make no proof. Therefore, the presented model for Raf activation by Ras binding remains untested and the authors wasted a good opportunity to show the validity of their model. A p-value of 0.496 generated by the program PISA is not indicative of a hydrophobic interface especially if the cutoff is 0.5.

Author Response

In their revised manuscript, the authors addressed some of
the points I raised in my previous critique. Unfortunately, they only
addressed points related to the presentation of the manuscript. The main
issue related to the validity of their model was not addressed. For example,
not a single mutant at the Ras dimer interface or to validate the molecular
dynamics results was generated as requested. Repeated arguments devoid of
experimental data to backup claims make no proof. Therefore, the presented
model for Raf activation by Ras binding remains untested and the authors
wasted a good opportunity to show the validity of their model.

Response: I responded to these critiques in my first revision. I have tried to make it as clear as possible that the goal here is not to prove the model, but to present an alternative model that still needs to be tested, as do the other currently existing models.

A p-value of 0.496 generated by the program PISA is not indicative of a hydrophobic
interface especially if the cutoff is 0.5.

Response: We modified the relevant sentence at the bottom of page 12 to recognize the fact that the p-value of 0.496 is practically at the cutoff. Rather than starting the sentence with “In support of a biologically-relevant dimer…” we now state “We have used the Protein Interfaces Surfaces and Assemblies (PISA) [49] website at the European Bioinformatics Institute to calculate the surface area buried in the KRas/CRaf-RBD_CRD dimer interface (PDB ID 6XI7) as 1005 Å2, with a DiG P-value of 0.496, just within the cutoff indicative of an interaction-specific interface unlikely to be an artifact of crystal packing”.